# Emerging fungal pathogen of an invasive grass: Implications for competition with native plant species

Amy E. Kendig[1]ᵒ*, Vida J. Svahnström[2]ᵒ, Ashish Adhikari[3], Philip F. Harmon[3], S. Luke Flory[1]

**1** Agronomy Department, University of Florida, Gainesville, Florida, United States of America, **2** School of Biology, University of St. Andrews, St. Andrews, Scotland, **3** Department of Plant Pathology, University of Florida, Gainesville, Florida, United States of America

ᵒ These authors contributed equally to this work.
* aekendig@gmail.com

**Data Availability Statement:** Data and code associated with this publication may be accessed through the Environmental Data Initiative Data Portal: https://doi.org/10.6073/pasta/

## Abstract

Infectious diseases and invasive species can be strong drivers of biological systems that may interact to shift plant community composition. For example, disease can modify resource competition between invasive and native species. Invasive species tend to interact with a diversity of native species, and it is unclear how native species differ in response to disease-mediated competition with invasive species. Here, we quantified the biomass responses of three native North American grass species (*Dichanthelium clandestinum*, *Elymus virginicus*, and *Eragrostis spectabilis)* to disease-mediated competition with the non-native invasive grass *Microstegium vimineum*. The foliar fungal pathogen *Bipolaris gigantea* has recently emerged in *Microstegium* populations, causing a leaf spot disease that reduces *Microstegium* biomass and seed production. In a greenhouse experiment, we examined the effects of *B. gigantea* inoculation on two components of competitive ability for each native species: growth in the absence of competition and biomass responses to increasing densities of *Microstegium*. *Bipolaris gigantea* inoculation affected each of the three native species in unique ways, by increasing (*Dichanthelium*), decreasing (*Elymus*), or not changing (*Eragrostis*) their growth in the absence of competition relative to mock inoculation. *Bipolaris gigantea* inoculation did not, however, affect *Microstegium* biomass or mediate the effect of *Microstegium* density on native plant biomass. Thus, *B. gigantea* had species-specific effects on native plant competition with *Microstegium* through species-specific biomass responses to *B. gigantea* inoculation, but not through modified responses to *Microstegium* density. Our results suggest that disease may uniquely modify competitive interactions between invasive and native plants for different native plant species.

c85303b29d66e7deb3387215a07015be Additional data is within the Supporting Information files.

**Funding:** The authors were supported by USDA award 2017-67013-26870 as part of the joint USDA-NSF-NIH Ecology and Evolution of Infectious Diseases program (https://www.usda.gov/). The funders had no role in study design, data collection and analysis, decision to publish, or preparation of the manuscript.

**Competing interests:** The authors have declared that no competing interests exist.

## Introduction

Both plant invasions and infectious diseases can affect natural plant communities by reducing plant diversity and biomass production [1–4]. Invasive species and disease outbreaks can co-occur in communities because the species are co-introduced, or because invasive species amplify disease transmission [5]. Invasive plants can negatively impact native plant communities through competition [6] and diseases may increase, decrease, or have no net effect on invasive plant impacts [7, 8]. The responses of both the invasive species and competing native species to infection can determine which of these outcomes occurs [9, 10]. For example, infected invasive plants are predicted to have lower competitive effects than uninfected invasive plants when native species have greater disease resistance or tolerance [11]. In empirical and theoretical tests of disease-mediated competition between a single native plant species and a single invasive plant species, disease has both increased [12, 13] and decreased [14, 15] impacts of the invasive species. However, the relevant guild of native species in natural communities is often diverse and species vary in their susceptibility to pathogen infection [16], making it unclear whether results from studies of disease-mediated competition with one native species apply to the broader native community.

A shared pathogen can create an antagonistic interaction between two or more host species (i.e., apparent competition), even in the absence of other forms of interaction, such as resource competition [17]. Increases in the abundance of one host species can decrease the fitness of another through transmission and the negative effects of disease [17]. Invasive species may enhance pathogen transmission to or disease impacts on native species [18, 19]. For example, high densities of invasive cheatgrass (*Bromus tectorum*) promoted infection of native plant seeds by a fungal pathogen [20]. Pathogen infection of native or invasive plants can reduce growth, reproduction, and survival [2, 21], as well as induce compensatory growth or reproduction [22, 23]. Because disease can influence the productivity and composition of natural plant communities [4], disease amplification by invasive plants could have potentially strong effects on native plant communities [24].

Disease can modify the effects of plants on shared resources and their responses to resource limitation [25]. For example, disease-induced reductions in total leaf area can decrease light interception [26], potentially increasing light availability to lower canopy levels and decreasing photosynthesis of infected plants. If disease disproportionately impacts invasive species, reductions in growth and resource uptake may release native species from competition, which was reported when a powdery mildew fungus infected invasive garlic mustard (*Alliaria petiolata*) [15]. In contrast, disease-induced fitness costs may reduce the competitive ability of native species, which is hypothesized to have promoted invasion of European grasses in California [12, 13]. Plant species can vary widely in their competitive ability [27, 28], susceptibility to infection [29, 30], and performance losses due to disease [23, 31]. Differential responses of native species to disease and competition with an invasive species could determine how the native plant community responds to disease-mediated competition.

*Microstegium vimineum* (Trin.) A. Camus (stilt grass, hereafter *Microstegium*) is an annual grass species native to Asia that was first recorded in the United States in 1919 [32]. *Microstegium* forms dense populations and litter layers in eastern and midwestern U.S. forest understories, suppressing herbaceous plants and tree seedlings [28, 33]. Over the last two decades, *Microstegium* populations have acquired fungal leaf spot diseases caused by species in the genus *Bipolaris* that reduce its biomass and seed production [29, 34]. *Bipolaris gigantea* (Heald & F.A. Wolf) B. Lane, Stricker, M.E. Sm., S.L. Flory & Harmon is a common pathogen of *Microstegium* [34] that causes zonate leaf spots with dark brown margins [35] and likely

disperses via wind and splashing [36], but with restricted distances due to its large spore size [35, 37].

*Dichanthelium clandestinum* L. Gould (syn. *Panicum clandestinum* L.; deer-tongue grass, hereafter *Dichanthelium*), *Elymus virginicus* L. (Virginia wild rye, hereafter *Elymus*), and *Eragrostis spectabilis* (Pursh) Steud (purple lovegrass, hereafter *Eragrostis*) are perennial grass species that are native to the U.S. and co-occur with *Microstegium* [29, 34]. *Bipolaris gigantea* can infect, produce lesions on, and produce spores on *Elymus virginicus* [35]. *Bipolaris gigantea* infections have been observed on *Dichanthelium clandestinum* in the field, but it may not be a competent host for spore production [37]. At least three species in the genus *Eragrostis* are susceptible to *B. gigantea* infection [37–39] and closely related plant species are more likely to share pathogens than distantly related species [30], suggesting that *Eragrostis spectabilis* may also be susceptible to *B. gigantea*. Infection by *B. gigantea* may be more likely when these grass species co-occur with *Microstegium* due to high infection rates in some *Microstegium* populations [34, 40]. *Bipolaris gigantea* may alter the competitive ability of *Microstegium* or the ability of these native species to compete with *Microstegium*.

Competitive ability depends on the fitness of species in the absence of competition and their responses to changes in intraspecific and interspecific competitor densities (i.e., competition coefficients) [41, 42]. Here we investigated how *B. gigantea* inoculation affected the competitive ability of the three native perennial grass species in a greenhouse experiment by measuring their competition-free biomass and responses to *Microstegium* density. We were uncertain about how *B. gigantea* would affect native species biomass, but acknowledged that a range of outcomes were possible given interspecific variation in host-pathogen interactions [29–31], including decreased, increased (e.g., through compensatory growth), and no change in biomass. We hypothesized that *B. gigantea* infection would reduce the negative effect of *Microstegium* density on native plant biomass because diseased *Microstegium* would be smaller [29, 34]. However, we also expected that disease-induced biomass reduction experienced by some native species would increase their sensitivity to *Microstegium* density.

## Materials and methods

### Greenhouse experiment

We performed the experiment in a greenhouse in Gainesville, FL, USA, from June 19, 2019 to September 12, 2019. We used *Microstegium* seeds collected from Big Oaks National Wildlife Refuge (BONWR) in Madison, IN, USA (38.9365, -85.4148) in 2015, *Elymus* and *Eragrostis* seeds purchased from Prairie Moon Nursery (Winona, MN, USA) in 2018, and *Dichanthelium* seeds purchased from Sheffield's Seed Company (Locke, NY, USA) in 2018. All seeds were stored at 4°C. Prior to the experiment, seeds of each species were planted in a greenhouse and seedlings developed no lesions, suggesting that lesions caused by *B. gigantea* inoculation were unlikely to be confused with lesions caused by potential seedborne pathogens. The potting mix used in the experiment (Jolly Gardener Pro-Line Custom Growing Mix) was autoclaved at 120–130°C for 30 minutes and all pots and trays were sprayed with 10% bleach solution (0.6% sodium hypochlorite) and rinsed with tap water after approximately five minutes to minimize risk of contamination by non-focal pathogens.

To quantify the effect of *Microstegium* competition on the native species, we used an additive competition experimental design [41, 43] with one individual of a native species surrounded by 0, 2, 10, 50, or 100 *Microstegium* plants per 1 L pot (Fig 1A and 1B). First, we sowed seeds for each native species into separate trays to germinate. Seven days later, we planted *Microstegium* seeds in 1 L pots according to their density treatment (50 and 100 seed numbers estimated by weight). The native species were transplanted from germination trays to

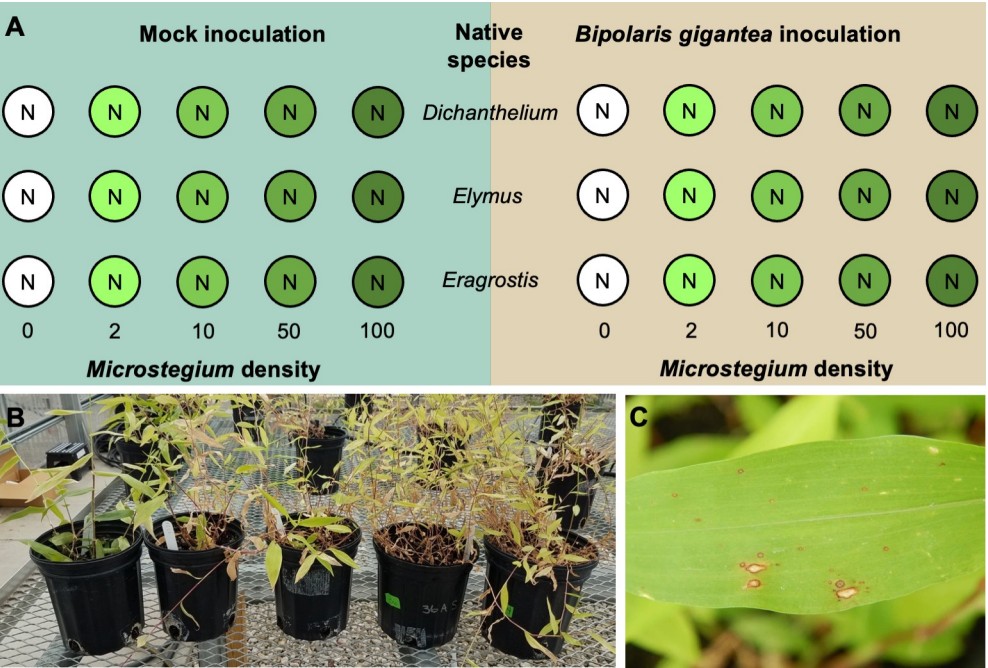

**Fig 1. Experimental methods.** (A) A diagram of the experimental design, (B) an example of the realized *Microstegium* density gradient (with *Dichanthelium* as the native species), and (C) an example of *Bipolaris*-like lesions on a *Microstegium* leaf from the experiment. Circles in A represent 1 L pots, with "N" indicating the central position of the native plant and the intensity of green shading indicating the *Microstegium* density (planted density values labelled below pots). Each represented pot was replicated four times.

the 1 L pots with *Microstegium* after growing in the greenhouse for 28 days. We chose native plant individuals that were similar in size (2 to 3 true leaves) to transplant into pots. The 15 plant combinations (each of the three native species with five *Microstegium* densities) were replicated eight times, half of which were inoculated with *B. gigantea* and half of which were mock inoculated with a control solution.

The pure culture of *B. gigantea* (BGLMS-1 in the collection of Dr. Philip Harmon, University of Florida) used in this research was originally isolated from *Microstegium* as part of a previous study and had been stored as previously described [35]. *Bipolaris gigantea* was revived from 4°C storage by placing colonized, 3 to 5 mm diameter, filter paper pieces on half-strength V8 media agar plates. Fungal colonies grew under 12 h day and night fluorescent light at 26°C for one week and were transferred to new half-strength V8 media agar plates. Conidia were harvested from fungal colonies by flooding plates with 10 ml of sterile deionized water with 0.1% Tween 20 (Sigma-Aldrich, St. Louis, MO, USA). The resulting conidia suspension was filtered through a layer of cheese cloth, and conidia were quantified with a Spencer Bright-Line hemocytometer (American Optical Company, Buffalo, NY, USA). The concentration of inoculum was adjusted to 15,000 conidia/ml and applied to plants with a Passche H-202S airbrush sprayer (Kenosha, WI, USA). Inoculations occurred six days after planting the native species with the *Microstegium*, and half of the pots were sprayed until runoff with the spore suspension while the other half were sprayed with the same volume of sterile deionized water with 0.1% Tween 20 (i.e., mock inoculation control). To encourage infection, we placed a paper towel wet with deionized water in each pot and sealed each pot with a transparent plastic bag secured with a rubber band. The plastic bags and paper towels were removed after seven days [44]. Plants were watered daily before and after they were contained in the plastic bags. Ten days

after bag removal, all plants were sprayed with Garden Safe insecticidal soap (Bridgeton, MO, USA) to help control aphids and thrips. Pots occupied two neighboring greenhouse benches and were haphazardly rearranged weekly to avoid confounding spatial positions with experimental treatments. *Bipolaris gigantea* isolations were in accordance with the United States Department of Agriculture Animal and Plant Health Inspection Service Plant Protection and Quarantine (USDA APHIS PPQ) permit no. PP526P-18-01688. Collections from BONWR were performed under a special use permit administered by the United States Fish and Wildlife Service.

## Data collection

To assess disease incidence (i.e., the percentage of leaves per pot with at least one lesion [45]) 14 days after inoculation, we recorded the number of *Microstegium* leaves with one or more *Bipolaris*-like lesions (Fig 1C) and the total number of leaves for three *Microstegium* plants per pot (or two plants for the pots with only two). No other types of lesions were observed on the plants. The number of leaves per plant were averaged within pots and multiplied by the total number of plants per pot, based on seeding rate, to estimate the total number of *Microstegium* leaves per pot. For native plants that received the *B. gigantea* inoculation treatment and had lesions, we counted the number of leaves with at least one lesion and the total number of leaves per plant. None of the plants in mock-inoculated pots had lesions with one exception: in one pot that contained *Dichanthelium* and 100 *Microstegium* plants, 46 *Microstegium* leaves had lesions. We removed this replicate from analyses.

We used these visual assessments of *Bipolaris*-like lesions to approximate *B. gigantea* infection of experimental plants. *Bipolaris gigantea* association with foliar lesions can be assessed by using microscopy to identify conidiophores on leaves after incubation [35]. The absence of *B. gigantea* conidiophores in lesions, however, does not confirm that it is not the causal agent. In leaf samples collected from BONWR in 2018 and 2019, 67% of *Microstegium* samples ($n = 238$) and 48% of *Elymus* samples ($n = 65$) that had *Bipolaris*-like lesions also had *B. gigantea* conidiophores identifiable by microscopy. In addition, 28% of *Microstegium* samples ($n = 29$) and 1% of *Elymus* samples ($n = 106$) that did not have *Bipolaris*-like lesions had *B. gigantea* conidiophores identifiable by microscopy (S1 Table). Therefore, *Bipolaris*-like lesions are commonly associated with *B. gigantea* infection and it is less common for leaves to be infected without lesions. Because we did not test leaves for infection with *B. gigantea* following inoculation, we present results in the context of the inoculation treatments rather than infection status.

To assess plant performance, we harvested the aboveground biomass of all pots on September 12, 2019 (51 days after inoculation), separated the native plants from the *Microstegium*, dried the biomass at 60˚C to constant mass, and weighed it. Biomass production can act as a proxy for perennial plant fitness [41]. While seed production is a more meaningful measure of annual plant fitness [41], *Microstegium* biomass is correlated with its seed production [46].

## Statistical analyses

To evaluate disease incidence on plants across the *Microstegium* density gradient, we fit a generalized linear regression to the estimated proportion of *Microstegium* leaves with lesions per pathogen-inoculated pot using *Microstegium* density (the number of *Microstegium* seeds added to each pot), native species identity, and their interaction as the explanatory variables. The model was fit with Bayesian statistical inference using the brm function in the brms package [47], an interface for Stan [48], in R version 3.5.2 [49]. The model contained three Markov chains with 6000 iterations each and a discarded burn-in period of 1000 iterations. We chose

prior distributions based on whether model variables could reasonably take on negative values in addition to positive values (Gaussian or Cauchy) or not (gamma or exponential). We chose parameters for prior distributions that reflected limited a priori information about variable values. We used a binomial response distribution (logit link) and a Gaussian distribution for the intercept and coefficient priors (location = 0, scale = 10). We calculated the mean and 95% highest posterior density interval (hereafter, "credible interval") of back-transformed (from logit to percentage) model estimates using the mean_hdi function in the tidybayes package [50]. There were too few native plant leaves with lesions to statistically analyze disease incidence, nevertheless, we present these results graphically to assess qualitative patterns.

To evaluate the effects of *Microstegium* density and *B. gigantea* inoculation on *Microstegium* performance, we fit a linear regression to *Microstegium* biomass:

$$biomass \sim native\ species \times inoculation \times (Microstegium\ density + Microstegium\ density^2).$$

This formulation allowed us to estimate quadratic relationships between *Microstegium* biomass and *Microstegium* density for each native species–inoculation treatment combination. We used a Gaussian response distribution, a Gaussian distribution for the intercept prior (location = 2, scale = 10) and the coefficient priors (location = 0, scale = 10), and a Cauchy distribution for the standard deviation prior (location = 0, scale = 1). Otherwise, the model was fit using the same methods described for disease incidence.

To evaluate the effects of inoculation treatment and *Microstegium* density on native plant biomass, we fit a Beverton-Holt function to native plant biomass:

$$bismass = \frac{b_0}{1 + \alpha \times Microstegium\ density}.$$

We fit this function to all of the native plant biomass data, estimating separate $b_0$ (biomass in the absence of competition) and $\alpha$ (biomass response to *Microstegium* density, i.e., competition coefficient) values for each native species–inoculation treatment combination. We used a Gaussian response distribution, a Gamma distribution for the $b_0$ prior (shape = 2, scale = 1), an exponential distribution for the $\alpha$ prior (rate = 0.5), and a Cauchy distribution for the standard deviation prior (location = 0, scale = 1). Otherwise, the model was fit using the same methods described for disease incidence. To evaluate differences in $b_0$ and $\alpha$ between treatments, we subtracted the estimate for one treatment from the other for each posterior sample ($n$ = 1500) and then calculated the mean and 95% credible intervals [50]. To assess model fits, we checked that the r-hat value for each parameter was equal to one, visually examined convergence of the three chains, and compared the observed data to simulated data from the posterior predictive distributions using the pp_check function [47]. We report a model coefficient as statistically significant if its 95% credible interval ("CI", i.e., 95% probability that this interval of the posterior distribution contains the true estimate value) omits zero [51, 52]. We used the tidyverse packages to clean data and create figures [53].

## Results

We observed *Bipolaris*-like lesions on *Microstegium* leaves in 94% of pots in which *Microstegium* was planted and inoculated. The average *Microstegium* disease incidence in low-density pots (i.e., two *Microstegium* plants) was 8% (95% CI: 6%–11%). The species identity of the native plant in low density pots did not significantly affect *Microstegium* disease incidence (Table 1). *Microstegium* disease incidence was constant across the *Microstegium* density gradient when *Elymus* was present (Fig 2B). However, *Microstegium* disease incidence decreased by four percentage points (95% CI: -7%–-2%) and five percentage points (95% CI: -7%–-3%)

**Table 1.** *Microstegium* disease incidence model.

| Coefficient | Estimate | Est. Error | Lower | Upper |
|---|---|---|---|---|
| **intercept** | **-2.33** | **0.13** | **-2.60** | **-2.07** |
| **density** | **-0.01** | **1.89E-03** | **-0.01** | **-3.70E-03** |
| *Elymus* | -0.24 | 0.18 | -0.59 | 0.12 |
| *Eragrostis* | -0.06 | 0.19 | -0.44 | 0.32 |
| **density:*Elymus*** | **0.01** | **2.46E-03** | **6.72E-04** | **0.01** |
| density:*Eragrostis* | -2.15E-03 | 2.73E-03 | -0.01 | 3.18E-03 |

Model-estimated parameters for generalized linear regression of the estimated proportion of *Microstegium* leaves with lesions from the *B. gigantea* inoculation treatment. The intercept is based on *Microstegium* grown with *Dichanthelium*. Estimate is the mean and Est. Error is the standard deviation of the posterior distribution. Lower and Upper are the lower and upper 95% credible intervals, respectively. Estimates with 95% credible intervals that exclude zero are in bold.

when 100 *Microstegium* were grown with *Dichanthelium* and *Eragrostis*, respectively, relative to pots with two *Microstegium* plants and the same native species (Fig 2A and 2C). The total number of leaves per pot increased across the *Microstegium* density gradient while the number of leaves with lesions increased more slowly or not at all (S1 Fig).

*Bipolaris gigantea* inoculation resulted in lesions on all three native plant species but only in some of the *Microstegium* density treatments (Fig 3). Lesions formed on 7 out of 20 *Elymus* plants (Fig 3B), but only 3 out of 20 plants for each of the other species (Fig 3A and 3C). Of the *Dichanthelium* and *Elymus* plants with lesions, higher *Microstegium* density tended to increase the percentage of leaves with lesions. For example, 17% of *Dichanthelium* leaves had lesions when grown with 100 *Microstegium* compared to 10% with 10 *Microstegium*. Similarly, 38% of *Elymus* leaves had lesions when grown with 100 *Microstegium* compared to 23% with 10 *Microstegium*.

*Bipolaris gigantea* inoculation did not significantly affect *Microstegium* biomass relative to the mock inoculation control (Table 2). In addition, *Microstegium* biomass was not significantly different among treatments with different native species (Table 2). *Microstegium* biomass increased with *Microstegium* density (0.16 g plant$^{-1}$, 95% CI: 0.05 g plant$^{-1}$–0.28 g plant$^{-1}$)

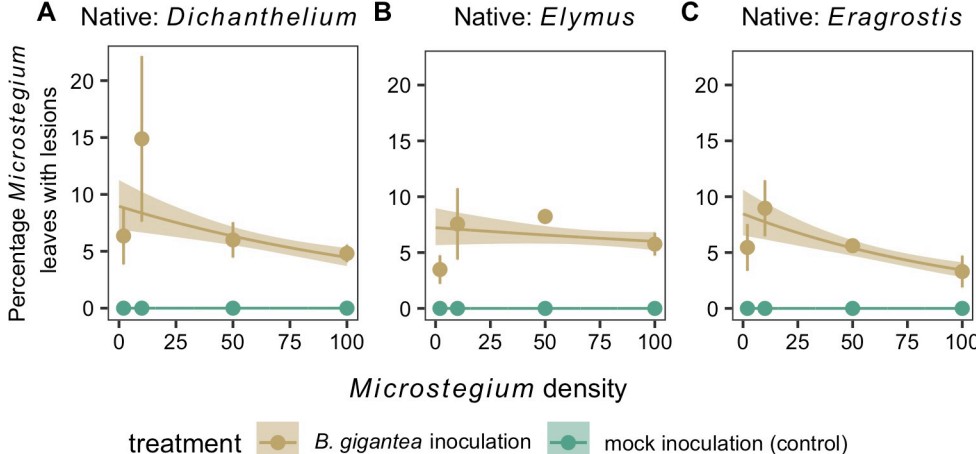

**Fig 2.** *Microstegium* disease incidence. The percentage of *Microstegium* leaves per pot with lesions following *B. gigantea* or mock inoculation across the *Microstegium* density gradient in the presence of (A) *Dichanthelium*, (B) *Elymus*, and (C) *Eragrostis*. Observations (points and error bars, mean ± 95% confidence intervals) and model fits (lines and shaded ribbons, mean ± 95% credible intervals) are shown.

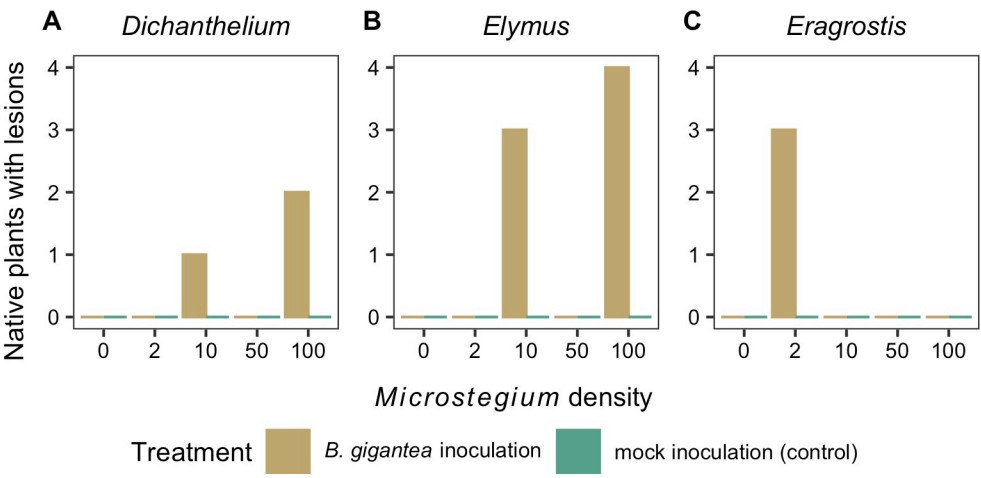

**Fig 3. Native plant disease.** The number of (A) *Dichanthelium*, (B) *Elymus*, and (C) *Eragrostis* plants with foliar lesions (out of four replicates) following *B. gigantea* or mock inoculation across the *Microstegium* density gradient.

and the relationship between biomass and density varied in shape, although not significantly, when grown with the three species: saturating at high densities when grown with *Dichanthelium* (Fig 4A), increasing nearly linearly when grown with *Elymus* (Fig 4B), and peaking at intermediate densities when grown with *Eragrostis* (Fig 4C).

In the absence of competition, the effects of *B. gigantea* inoculation on native plant biomass ($b_0$) depended on the native plant species (Table 3), increasing *Dichanthelium* biomass by 1.19

**Table 2. *Microstegium* biomass model.**

| Coefficient | Estimate | Est. Error | Lower | Upper |
|---|---|---|---|---|
| intercept | 1.77 | 0.92 | -0.04 | 3.58 |
| **density** | **0.16** | **0.06** | **0.05** | **0.28** |
| *B. gigantea* inoculation | 0.56 | 1.30 | -2.02 | 3.09 |
| *Elymus* | 2.54 | 1.32 | -0.01 | 5.13 |
| *Eragrostis* | -0.63 | 1.33 | -3.30 | 2.00 |
| density$^2$ | -9.76E-04 | 5.64E-04 | -2.08E-03 | 1.36E-04 |
| density:*B. gigantea* inoculation | -0.05 | 0.08 | -0.20 | 0.11 |
| density:*Elymus* | -0.11 | 0.08 | -0.27 | 0.05 |
| density:*Eragrostis* | -1.00E-03 | 0.08 | -0.16 | 0.16 |
| *B. gigantea* inoculation:*Elymus* | -1.32 | 1.85 | -4.98 | 2.32 |
| *B. gigantea* inoculation:*Eragrostis* | 0.15 | 1.86 | -3.48 | 3.85 |
| *B. gigantea* inoculation:density$^2$ | 1.08E-04 | 7.84E-04 | -1.44E-03 | 1.61E-03 |
| *Elymus*:density$^2$ | 7.90E-04 | 7.83E-04 | -7.75E-04 | 2.30E-03 |
| *Eragrostis*:density$^2$ | -2.93E-04 | 8.02E-04 | -1.86E-03 | 1.26E-03 |
| density:*B. gigantea* inoculation:*Elymus* | 0.05 | 0.11 | -0.18 | 0.27 |
| density:*B. gigantea* inoculation:*Eragrostis* | 0.06 | 0.11 | -0.17 | 0.28 |
| *B. gigantea* inoculation:*Elymus*:density$^2$ | -1.14E-04 | 1.10E-03 | -2.25E-03 | 2.04E-03 |
| *B. gigantea* inoculation:*Eragrostis*:density$^2$ | -4.24E-04 | 1.11E-03 | -2.60E-03 | 1.81E-03 |

Model-estimated parameters for linear regression model of *Microstegium* biomass. The intercept is based on *Microstegium* grown with *Dichanthelium*. Estimate is the mean and Est. Error is the standard deviation of the posterior distribution. Lower and Upper are the lower and upper 95% credible intervals, respectively. Estimates with 95% credible intervals that exclude zero are in bold.

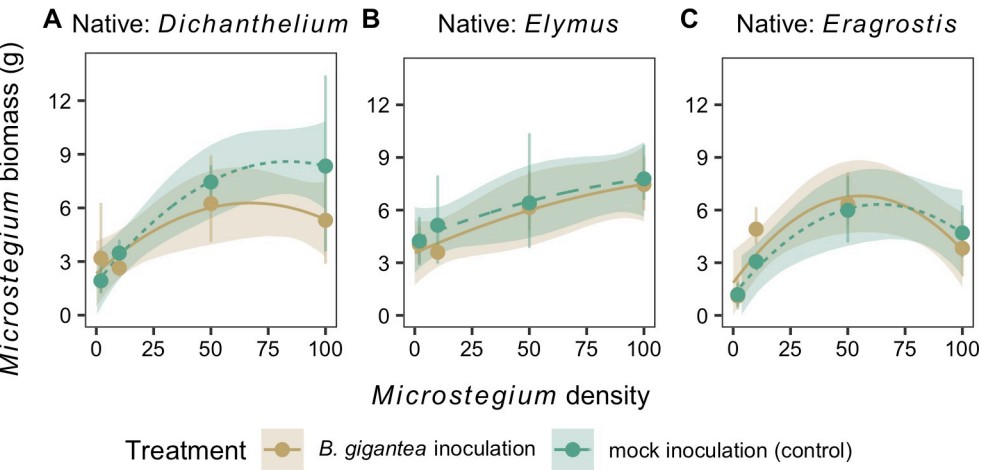

**Fig 4. *Microstegium* biomass.** The biomass of *Microstegium* following *B. gigantea* or mock inoculation across the *Microstegium* density gradient in the presence of (A) *Dichanthelium*, (B) *Elymus*, and (C) *Eragrostis*. Observations (points and error bars, mean ± 95% confidence intervals) and model fits (lines and shaded ribbons, mean ± 95% credible intervals) are shown.

g (95% CI: 0.82 g–1.59 g; Fig 5A), decreasing *Elymus* biomass by 0.60 g (95% CI: -0.99 g–-0.22 g; Fig 5B), and having no significant effect on *Eragrostis* biomass (estimated change: -0.17 g, 95% CI: -0.55 g–0.20 g; Fig 5C) relative to the mock inoculation control. The effect of *Microstegium* density on native biomass ($\alpha$) was consistent across the native species (Table 3), with an average value of 2.08 plant$^{-1}$ (95% CI: 0.61 plant$^{-1}$–6.28 plant$^{-1}$) in the mock inoculation treatment. There were no significant effects of *B. gigantea* inoculation on the responses of the three native species to increases in *Microstegium* density (average *B. gigantea* inoculation effect: -0.57 plant$^{-1}$, 95% CI: -5.37 plant$^{-1}$–3.48 plant$^{-1}$).

## Discussion

We evaluated how inoculation with the emerging fungal pathogen *B. gigantea* affected the biomasses of three native species in competition with the invasive plant *Microstegium*. *Bipolaris*

**Table 3. Native plant biomass model.**

| Parameter | Species | Inoculation | Estimate | Est. Error | Lower | Upper |
|---|---|---|---|---|---|---|
| $b_0$ | *Dichanthelium* | mock | 2.10 | 0.14 | 1.82 | 2.38 |
| $b_0$ | *Dichanthelium* | *B. gigantea* | 3.30 | 0.14 | 3.02 | 3.57 |
| $b_0$ | *Elymus* | mock | 1.89 | 0.14 | 1.61 | 2.17 |
| $b_0$ | *Elymus* | *B. gigantea* | 1.29 | 0.14 | 1.03 | 1.56 |
| $b_0$ | *Eragrostis* | mock | 1.02 | 0.14 | 0.75 | 1.29 |
| $b_0$ | *Eragrostis* | *B. gigantea* | 0.85 | 0.14 | 0.58 | 1.12 |
| $\alpha$ | *Dichanthelium* | mock | 1.98 | 0.89 | 0.93 | 4.27 |
| $\alpha$ | *Dichanthelium* | *B. gigantea* | 2.01 | 0.57 | 1.21 | 3.39 |
| $\alpha$ | *Elymus* | mock | 3.49 | 1.74 | 1.40 | 7.99 |
| $\alpha$ | *Elymus* | *B. gigantea* | 2.49 | 1.55 | 0.78 | 6.65 |
| $\alpha$ | *Eragrostis* | mock | 2.94 | 1.91 | 0.76 | 7.91 |
| $\alpha$ | *Eragrostis* | *B. gigantea* | 2.20 | 1.70 | 0.44 | 6.79 |

Model-estimated parameters for Beverton-Holt models of native plant biomass. Estimate is the mean and Est. Error is the standard deviation of the posterior distribution. Lower and Upper are the lower and upper 95% credible intervals, respectively.

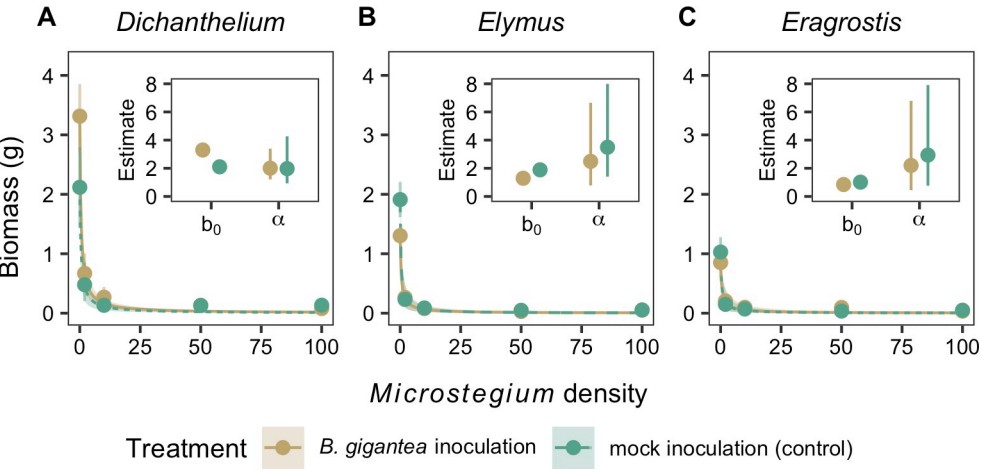

**Fig 5. Native plant biomass.** The biomass of (A) *Dichanthelium*, (B) *Elymus*, and (C) *Eragrostis* across the *Microstegium* density gradient following *B. gigantea* or mock inoculation. Main plots show observations (points and error bars, mean ± 95% confidence intervals) and model fits (lines and shaded ribbons, mean ± 95% credible intervals). Inset plots show the model-estimated biomass in the absence of competition ($b_0$) and the biomass responses to *Microstegium* density ($\alpha$) (mean ± 95% credible intervals).

*gigantea* inoculation did not significantly affect *Microstegium* biomass relative to the mock inoculation control and it had contrasting effects on the native plant species by increasing *Dichanthelium* biomass, decreasing *Elymus* biomass, and having no effect on *Eragrostis* biomass. The negative effect of *Microstegium* density on biomass for each of the native species was the same whether plants were inoculated with *B. gigantea* or the control. These results suggest that *B. gigantea* may differentially affect native plant species in competition with *Microstegium* through species-specific responses to *B. gigantea* exposure.

## Disease-mediated competition

Studies on disease-mediated competition between invasive and native plant species indicate that infection of invasive plants can contribute to either native plant persistence or recovery [14, 15, 34, 54]. However, in our experiment, inoculation with *Bipolaris* did not modify the effect of *Microstegium* density on the three native species relative to the mock inoculation control, suggesting that competitive effects of *Microstegium* on native species are likely to be consistent in the presence or absence of low levels of disease incidence. Our experimental design captured two components of native species' competitive ability: their growth in the absence of competition and their biomass responses to changes in *Microstegium* density (i.e., interspecific competition coefficients) [41, 42]. To better characterize the competitive ability of the native species, it is also necessary to estimate their biomass responses to changes in their own density (i.e., intraspecific competition coefficients) [41, 42]. Because *B. gigantea* inoculation had unique effects on the growth of the three native species, it may also uniquely affect their per capita impacts on competitor growth, altering their intraspecific competition coefficients. If, however, *B. gigantea* does not affect the intraspecific competition coefficients of the native species, we would expect low levels of *B. gigantea* exposure to increase the competitive ability of *Dichanthelium* in interactions with *Microstegium*, decrease the competitive ability of *Elymus*, and to have no effect on the competitive ability of *Eragrostis*, potentially leading to shifts in the relative abundances of native species. Predicting the long-term outcome of disease impacts on native–invasive interactions requires also understanding the invasive species' competitive ability and niche overlap between the native and invasive species (e.g., overlapping resource

requirements or natural enemies) [25, 42, 55]. Because the native species are perennial and the invasive species is an annual, studies that examine the effects of disease on aspects of plant fitness other than biomass, for example, annual seed survival and perennial adult survival, are necessary for characterizing the comprehensive impacts of disease on native–invasive plant competition [41, 55].

Our experimental methods and conditions may have limited the levels of *B. gigantea* leaf spot disease (see Limitations section), and there were no effects of *B. gigantea* inoculation on *Microstegium* biomass, likely leading to limited changes in *Microstegium* resource acquisition. Therefore, it is crucial to explore disease-mediated competitive effects of *Microstegium* in the field or with methods that may result in disease incidence approaching levels observed in the field. For example, disease incidence on *Microstegium* decreased as *Microstegium* densities increased, which is likely because the single *B. gigantea* inoculation infected a relatively constant number of leaves regardless of *Microstegium* density, leading to lower percentages of leaves with lesions at higher densities. In contrast, higher plant densities in the field may promote higher disease incidence and greater inoculum production [56, 57], in which case disease may have stronger impacts on *Microstegium* competition than what we observed in the experiment. Greater *B. gigantea* inoculum levels and multiple disease cycles may also have larger effects on native plant responses to competition, for example, through reduced ability to capture resources [58]. Nonetheless, the impacts of *B. gigantea* on *Microstegium* and native plant competition may simply be minor, as has been demonstrated for disease effects on cheatgrass competition [14] and herbivory effects on Amur honeysuckle (*Lonicera maackii*) competition with native species [59]. In that case, competitive effects of the invasive plant on native species are likely to overshadow the effects of disease, which may be common across plant communities [60].

## Consequences of *B. gigantea* inoculation

It is likely that pathogen amplification by invasive plants has distinct effects on different native species [20, 31]. The three native species in our experiment showed unique biomass responses to *B. gigantea* inoculation in the absence of *Microstegium* competition. The range of *B. gigantea* inoculation effects on biomass, from negative (on *Elymus*) to positive (on *Dichanthelium*) is consistent with the theory that plant-microbe interactions can vary from mutualism to parasitism depending on context, such as environmental conditions and host identity [61, 62]. For example, infection with Cucumber mosaic virus increased the biomass and seed weight of one *Arabidopsis thaliana* genotype while it reduced the biomass and seed weight of another genotype relative to a mock inoculation control [63]. Our results suggest that *B. gigantea* could increase aboveground growth for some host species (e.g., through compensatory growth or re-allocation of resources [22, 63]) and suppress the aboveground growth of other species. Studies encompassing a broader range of environmental conditions and host diversity are needed to better predict when *B. gigantea* will have positive, negative, or neutral effects on host biomass and other traits. In studies of soil microbes, the effects of microbial inoculations on plant growth can predict plant species relative abundances in the field [64, 65]. However, whether disease-induced changes in growth of plant species, as observed in our study, are sufficient to shift plant community structure is an important area of future research [25].

Interestingly, no fungal lesions were observed on the three native plant species in the absence of competition despite seven days of incubation inside plastic bags. The absence of visible symptoms, however, does not necessarily indicate a lack of infection. For example, some fungi are asymptomatic endophytes of invasive Crofton weed (*Ageratina adenophora*) but cause visible leaf spots on co-occurring plant species [66]. Fungal lesions were observed on

some native plants grown with *Microstegium*, perhaps because *Microstegium* biomass altered the microclimate of the pots (e.g., increased humidity), which can be more suitable for lesion formation [56]. In addition, transmission from *Microstegium* to native plants may have maintained infections. Indeed, *B. gigantea* transmission from more competent host species to less competent host species has been inferred from field observations [37]. If *Microstegium* biomass amplifies *B. gigantea* incidence on native species in the field, *B. gigantea* could drive apparent competition between *Microstegium* and species negatively affected by *B. gigantea* [17].

## Limitations

Experimentally suppressing *Bipolaris* infection using fungicide in the field increased *Microstegium* biomass by 33–39% [29, 34], suggesting substantial effects of severe disease symptom development. However, despite using a pathogenic *Bipolaris* isolate in our experiment [35], inoculation caused low levels of disease incidence (relative to approximately 40% of leaves with lesions documented in the field, S2 Table), which had no effect on *Microstegium* biomass relative to the mock inoculation control. While such results could be explained by *Microstegium* tolerance or compensatory growth [22, 23], it is more parsimonious to assume that *B. gigantea* exposure was below levels experienced in the field. Pathogen transmission and disease incidence depend on the favorability and duration of environmental conditions and the inoculum load [67, 68]. Our experiment relied on a single inoculation and extended incubation; however, field conditions that result in cycles of leaf wetness events (e.g., dew or precipitation) can enhance fungal infection [67] and promote multiple disease cycles throughout the growing season. The concentration of *B. gigantea* conidia in our experimental inoculations was limited by the number of conidia we could harvest from agar plates in the lab and was relatively low (15,000 conidia/ml compared to e.g., $10^5$ conidia/ml [68]). While the conditions for leaf wetness and conidia suspension concentration likely limited the possible extent of disease incidence, they may reflect initial disease dynamics in the field, which is consistent with the age of plants we used in the experiment.

Although we did not test plants for infection with *B. gigantea* infection following the experiment, our results indicate that observed lesions and biomass effects were likely due to the *B. gigantea* inoculation treatment. Only a single pot in the mock inoculation treatment exhibited disease symptoms while 94% of *B. gigantea*–inoculated pots with *Microstegium* ($n$ = 48) displayed disease symptoms. In addition, a relatively high number of leaves in the one mock-inoculated pot had lesions. These two results suggest that the pot was inadvertently inoculated when treatments were applied. An alternative explanation is that the one mock-inoculated pot was contaminated (e.g., seeds harbored pathogenic fungi or external contaminants were introduced). While either is a possibility, the latter does not explain the much higher percentage of inoculated pots with lesions relative to mock-inoculated pots. In addition, *B. gigantea* can co-occur with other pathogens in the field (S3 Table). Therefore, infection of plants in our experiment that is not confounded with the inoculation treatment does not negate our ability to evaluate the effects of the inoculated *B. gigantea* strain on competition between native plants and *Microstegium*. Future efforts that aim to better characterize host-pathogen interactions between *Microstegium* or native plant species and *Bipolaris* fungi could confirm infection by attempting to re-isolate the fungus after inoculation.

## Conclusions

We used a greenhouse experiment to demonstrate that inoculation with a fungal leaf spot pathogen that has accumulated on a widespread invasive grass had unique effects on the growth of native species but did not modify biomass responses of native species to

*Microstegium* density. Complementary experiments in the field could help determine whether these findings are consistent across other native species and when disease pressure is higher. Transmission of *B. gigantea* may depend on *Microstegium* densities, potentially creating feedbacks between infection and density, which we controlled for in our experiment. The competitive effects of native plant species on *Microstegium* in the presence and absence of disease also may be important for understanding long-term community dynamics [8]. The emergence of infectious diseases in invaded plant communities may lead to natural biological control of the invasive species [7], exacerbated effects of invasion if the pathogen negatively impacts native species [12], or there may be no effect of disease [8]. Altogether, our study suggests that low levels of disease caused by *B. gigantea* may have unique effects on native species but are unlikely to modify the large negative impact of invasive *Microstegium* density on native species.

## Supporting information

**S1 Fig. *Microstegium* leaves.** The estimated number of *Microstegium* leaves with lesions across the *Microstegium* density gradient following *B. gigantea* inoculation when grown in the presence of (A) *Dichanthelium*, (B) *Elymus*, and (C) *Eragrostis* (mean ± 95% confidence intervals). All leaves with lesions were counted and the total number leaves per pot were estimated by counting the number of leaves on up to three plants per pot.
(TIF)

**S1 Table. *Bipolaris gigantea* identification on field-collected leaves.** Raw data collected from an experiment at BONWR in which fungicide or water (control) were added to plots with ten planting treatments. Leaves of *Microstegium vimineum* (Mv) and *Elymus virginicus* (Ev) were assessed for visible eyespots and *B. gigantea* conidiophores using microscopy.
(DOCX)

**S2 Table. *Microstegium vimineum* infection incidence in the field.** Raw data collected from *Microstegium* in an experiment at BONWR in which the total number of leaves per stem and the number of leaves with at least two foliar lesions per stem were recorded. The plots included were sprayed monthly with water a control for fungicide (not included).
(DOCX)

**S3 Table. Fungi identification on field-collected leaves.** Raw data collected from an experiment at BONWR in which fungicide or water (control) were added to plots with ten planting treatments. Leaves of *Microstegium vimineum* (Mv) and *Elymus virginicus* (Ev) were assessed for *B. gigantea* conidiophores using microscopy. *Bipolaris gigantea* was isolated, as well as some co-occurring fungi, including *Pyricularia* spp., *Bipolaris* spp. other than *B. gigantea*, and *Curvularia* spp. Leaves were collected in late August of 2018.
(DOCX)

## Acknowledgments

We would like to thank Liliana Benitez, Zobia Chanda, Laney Davidson, Zadok Jollie, and Shannon Regan for assistance with the experiment, Joe Robb for research guidance at Big Oaks National Wildlife Refuge, Simon Riley for assistance with the statistical analysis, and Brett Lane, Erica Goss, Robert Holt, Michael Barfield, Nicholas Kortessis, Margaret Simon, Christopher Wojan, and Keith Clay for discussions about the experiment and manuscript.

## Author Contributions

**Conceptualization:** Amy E. Kendig, Vida J. Svahnström, Philip F. Harmon, S. Luke Flory.

**Data curation:** Amy E. Kendig.

**Formal analysis:** Amy E. Kendig, Vida J. Svahnström.

**Funding acquisition:** Philip F. Harmon, S. Luke Flory.

**Investigation:** Amy E. Kendig, Vida J. Svahnström, Ashish Adhikari.

**Methodology:** Amy E. Kendig, Vida J. Svahnström, Ashish Adhikari, Philip F. Harmon, S. Luke Flory.

**Project administration:** Vida J. Svahnström.

**Software:** Amy E. Kendig.

**Supervision:** Amy E. Kendig, Philip F. Harmon, S. Luke Flory.

**Visualization:** Amy E. Kendig.

**Writing – original draft:** Amy E. Kendig.

**Writing – review & editing:** Vida J. Svahnström, Ashish Adhikari, Philip F. Harmon, S. Luke Flory.

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
