## [Decision Letter · Decision Letter 0]

23 Oct 2020

PONE-D-20-23926

Emerging fungal pathogen on an invasive grass differentially affects native species

PLOS ONE

Dear Dr. Kendig,

Thank you for submitting your manuscript to PLOS ONE. After careful consideration, we feel that it has merit but does not fully meet PLOS ONE’s publication criteria as it currently stands. Therefore, we invite you to submit a revised version of the manuscript that addresses the points raised during the review process.

I apologize for the delay in returning my decision to you. The reviews are mixed, with two reviewers recommending rejection and one recommending minor revision. I deeply respect and agree with all the reviewer's concerns about this paper, but I am willing to reconsider a revised version of the manuscript if the serious flaws noted by all three reviewers can be satisfactorily addressed. However, a revised version that does not adequately address their concerns will be rejected. Serious concerns to be addressed include, but are no limited to, questions about disease incidence/ severity, major issues regarding the methodology used, and a need to expand upon the inoculation procedure.

We look forward to receiving your revised manuscript.

Kind regards,

Richard A Wilson

Academic Editor

PLOS ONE

Journal Requirements:

2. In your Methods section, please provide additional location information of the collection sites, including geographic coordinates for the data set if available.

Reviewers' comments:

Reviewer's Responses to Questions

**Comments to the Author**

1. Is the manuscript technically sound, and do the data support the conclusions?

Reviewer #1: No

Reviewer #2: No

Reviewer #3: Yes

2. Has the statistical analysis been performed appropriately and rigorously? 

Reviewer #1: Yes

Reviewer #2: Yes

Reviewer #3: Yes

3. Have the authors made all data underlying the findings in their manuscript fully available?

Reviewer #1: Yes

Reviewer #2: Yes

Reviewer #3: No

4. Is the manuscript presented in an intelligible fashion and written in standard English?

Reviewer #1: Yes

Reviewer #2: Yes

Reviewer #3: Yes

5. Review Comments to the Author

Reviewer #1: The aim of this study was to determine how the pathogen (Bipolaris) and invasive Microstegium impacted competition among native and invasive plant.

Major comment:

It seems very strange that native plants would only develop symptoms in the presence of invasive Microstegium. This deserve some explanation. How does an invasive plant make native plants more susceptible to pathogen?

According to the authors, L177: ‘There were too few native plants with lesions. With that being said, how can the authors speculate the following at Line 268: ‘These results suggest that disease caused by Bipolaris may alter the community composition of native species. This claim seems farfetched and unsupported, since ‘there were too few native plants with lesions. The authors report disease incidence at Line 209 -213, and then provide an explanation that is not intuitive. The results are reported as ‘disease incidence’, but then authors explain disease severity. Disease incidence is should be per plant/individual. Disease severity should be per leaves.

It is safe to assume that the authors did not perform preliminary studies to determine if the pathogen could infect the native plants used in this study. L89 -93, the author explicitly mentions that because the native plants occur with the invasive, they assumed the natives were susceptible. The study would have actually benefited from screening, testing, and using susceptible natives. This is an interesting pilot study. An interesting result that might be worthy of explaining is the different outcome that Bipolaris had on 3 different native plants.

According to table S1, Microstegium disease incidence decreased as microstegium density increased. As mentioned before the explanation needs more insight.

As for the results section, the authors do not provide a P-values, but report significance via supplementary table, where 95% CI does not intersect zero. Truthfully, this is my first time reading a manuscript without a reported p value. The readers will find it odd that the results section does not mention a p value. values need to be included

 

Minor Comments

Line 21: After mentioning ‘North American grass species. I recommend adding a parenthesis and naming the native grass species.

Line 22: Hard stop, add period after Bipolaris gigantea. Now make a new sentence.

i.e This foliar pathogen has recently emerged in populations of the invasive grass Microstegium vimineum, causing leaf spot disease

Line 28: Remove: “inoculation treatment”. Replace with “pathogen outcome”

Line 29: Remove: “suggesting that disease had no indirect effects through altered competition’

ADD: “suggesting that pathogen outcome did not alter competition”:

Line 38: Replace “afflict” with ‘perturb’

Line 41 – 42: This is unclear because plant invasions are a spatiotemporal process that stems from (1) transport, (2) colonization, (3) establishment, and (4) landscape spread(Theoharides and Dukes, 2007), but It appears as if the author is using plant introduction synonymous with plant invasion. How is community structure being maintained, if an invasion suggests altered community structure

Line 43: This is unclear: “suppressing native species that would either driver on its own”.

Line 44: This sentence is unclear. Perhaps avoid starting a sentence with a conjunction (i.e “Which”), and it will be easier for the reader to understand

Line 45 – 47: This line can be more precise. Also, it seems untimely because at Line 41 -44 an example is given of ‘disease and plant invasion’. Line 45 – 47 is saying the same thing again. Only difference is in Line 41 -44 the reader is told the potential outcome of ‘disease and plant invasion’, and in Line 45 – 47 the outcome is vague but with more references.

Line 62: Replace: “occurred” with “was reported”

Line 70: Formally introduce the pathogen. Mentioning that an anonymous pathogen suppresses growth and reproduction and providing two reference seems to be a stating the obvious. In general, there are some references in this section, without the thought or example being fully defines

Line 92 - 93: Pathogen is formally mentioned, but there isn’t any mention of symptoms, or how it spreads. Also, the authors mention that the native grass species maybe to susceptible to Bipolaris, but due to the nature of this study, it seems a preliminary test to confirm susceptibility would have been helpful

L209 -213: A disease incidence is presence or absence of disease on a particular plant. However, the author seems to report disease incidence, and then gives a disease severity interpretation (i.e leaves with lesions)

L211 -213: Interpretation should be explained in the discussion

THEOHARIDES, K. A. & DUKES, J. S. 2007. Plant invasion across space and time: factors affecting nonindigenous species success during four stages of invasion. New phytologist, 176, 256-273.

Reviewer #2: Manuscirpt by Kendig et al investigates the impact of fungal pathogen Bipolaris on one invasive grass Microstegium vimineum in North America as well as on three native grasses, Dichanthelium clandestinum Elymus virginicus, and Eragrostis spectabilis. In addition their aim was to test how pathogen impacts on the competition between the plant species. The experimental design includes the three separate native grass species treatments alone and in different densities of invasive grass and Bipolaris inoculation and mock inoculations in greenhouse. The authors find that the pathogen has minor effect on the biomass of the native grasses in the absence of the invasive grass but the invasive grass has overall larger negative impact on the native grasses than the pathogen. However, there are major methodological issues that must be addressed before being able to evaluate the manuscript further:

Main problems: First, the authors do not soundly establish that the three Native grasses are susceptible for Bipolaris. This should be done by either referencing previous literature that explicitly shows that, in ref 24 Flory et al 2014 show that Elymus is susceptible to some strains of Bipolaris but not all of them. I did not find the other two grass species in the refs 24, 27, 32, 33. Or, alternatively by performing a laboratory trial on with the strain in concern and with all the native species and then using the strain in the experiment. Or, by taking the leaf samples from the plants used in the experiment, performing microscopy, growing the pathogen isolate on growth media plates and identification from them, or by PCR from the infected leaf tissue. None of these approaches is currently taken.

Second, the authors have not discussed the possibility that the symptoms in the study plants are caused by another pathogen, potentially from seed (again, lack of any detection methods in the experiment).

Third, Bipolaris infection found in the pathogen treated and mock inoculated plants (l. 158-160) may be seedborne. This should have be established by using pathogen free seed, testing all plant material after germination prior experiment for Bipolaris by PCR, or by testing mock inoculated plants with PCR in the end of the experiment.

Reviewer #3: The research article ‘Emerging fungal pathogen on an invasive grass differentially affects native species’ documents a well-planned experimental examination of the effects on three native grasses (Dichanthelium, Elymus and Eragrotis) of an invasive grass, Microstegium vimineum, in the presence of a fungal pathogen, Bipolaris gigantea. The manuscript is well written. Results show that the density of the invasive grass has effects on the native grasses that overwhelms any effect of the presence of the pathogen on either the invasive grass itself or the native grasses. Some differential effects of the fungus on the native grasses is observed, however infection patterns across individuals suggest that the inoculation protocol failed at some level – indeed the authors concede that conidial concentrations of the inoculum were low, perhaps by several orders of magnitude. I believe more needs to be said about the efficacy of the inoculation procedure. A more thorough discussion concerning the differential direct effects of the fungus on the native grasses that were infected would be welcome.

Detailed comments:

Line 38. ‘affect’ not ‘afflict’

Line 88. infections ‘with’ species not ‘from’

Line 101. plastics ‘bags’ not ‘bag’

Line 138. ‘hemocytometer’ (or haemocytometer) not ‘hemacytometer’

Line 165. incidence ‘on’ plants not ‘experienced by’

Line 239. (Fig. 4C) not (Fig. 4B)

Line 301. ‘do not’ rather than ‘don’t’

Line 308. sufficient to ‘shift plant’ community not ‘plant shift’

6. PLOS authors have the option to publish the peer review history of their article (what does this mean?). If published, this will include your full peer review and any attached files.

Reviewer #1: No

Reviewer #2: No

Reviewer #3: No

---

## [Author Response · Author response to Decision Letter 0]

8 Dec 2020

Please see the uploaded cover letter (before manuscript text) for our responses to editor and reviewer comments.

---

## [Decision Letter · Decision Letter 1]

6 Jan 2021

PONE-D-20-23926R1

Emerging fungal pathogen differentially affects three native plant species that compete with an invasive grass

PLOS ONE

Dear Dr. Kendig,

Thank you for submitting your manuscript to PLOS ONE. After careful consideration, we feel that it has merit but does not fully meet PLOS ONE’s publication criteria as it currently stands. Therefore, we invite you to submit a revised version of the manuscript that addresses the points raised during the review process.

Reviewer 2 and 3 felt the paper was much improved and have recommended accept. However, reviewer 1 was not satisfied and recommended rejection. The main sticking point is still the issue with Bipolaris. I wonder if it is possible to recover Bipolaris from infected lesions, i.e. perform Koch's postulates on a small number of representative infected leaves? Or, perhaps it is possible to artificially infect some plants with Bipolaris spores? Otherwise, the authors will need to better convince reviewer 1of the merits of the study as it stands.

We look forward to receiving your revised manuscript.

Kind regards,

Richard A Wilson

Academic Editor

PLOS ONE

Reviewers' comments:

Reviewer's Responses to Questions

**Comments to the Author**

1. If the authors have adequately addressed your comments raised in a previous round of review and you feel that this manuscript is now acceptable for publication, you may indicate that here to bypass the “Comments to the Author” section, enter your conflict of interest statement in the “Confidential to Editor” section, and submit your "Accept" recommendation.

Reviewer #1: (No Response)

Reviewer #2: All comments have been addressed

Reviewer #3: All comments have been addressed

2. Is the manuscript technically sound, and do the data support the conclusions?

Reviewer #1: No

Reviewer #2: Yes

Reviewer #3: (No Response)

3. Has the statistical analysis been performed appropriately and rigorously? 

Reviewer #1: Yes

Reviewer #2: Yes

Reviewer #3: (No Response)

4. Have the authors made all data underlying the findings in their manuscript fully available?

Reviewer #1: Yes

Reviewer #2: Yes

Reviewer #3: (No Response)

5. Is the manuscript presented in an intelligible fashion and written in standard English?

Reviewer #1: Yes

Reviewer #2: Yes

Reviewer #3: (No Response)

6. Review Comments to the Author

Reviewer #1: Major Comments

In this study, the authors address plant competition and plant pathogen infection. However, if the authors truly believe their study can stand without confirming that the native plants are susceptible to Bipolaris, then perhaps the authors should omit all text about Bipolaris infecting native plants. Currently, Line 276 says “Bipolaris gigantea inoculation resulted in lesions on all three native plant species, but only in the presence of Microstegium”. This statement cannot be made with confidence, the authors do not know if those lesions were caused by Bipolaris. Similar statements appear throughout the text (i.e L 333), and raises concern.

I acknowledge the authors explanation that the invasive competitor may have increased humidity (L413), which may have influenced disease symptoms on native plants, but it still seems like a bit of stretch. The authors should have confirmed that the pathogen can infect native plants, as part of a preliminary study or experimental design. This careful approach, would have allowed the authors to determine if humidity was a factor affecting susceptibility among native plants. Humidity could have easily been manipulated by confining the plant’s air-space with a plastic dome cover, or an autoclave bag, while exposing plants to Bipolaris. This approach is common in plant pathology cases that address foliar pathogens that infect via stomata. As it stands, the authors can only speculate if humidity was a factor.

In conclusion, the mere fact that the authors did not determine if Bipolaris can infect the native plants raises concern. It is possible that the lesions may not be from Bipolaris. It may be possible the invasive carries a seedborne endophyte, that is asymptomatic in the invasive, but when transmitted, causes symptoms on the natives, and this endophyte may not be Bipolaris.

Reviewer #2: (No Response)

Reviewer #3: (No Response)

7. PLOS authors have the option to publish the peer review history of their article (what does this mean?). If published, this will include your full peer review and any attached files.

Reviewer #1: No

Reviewer #2: No

Reviewer #3: No

---

## [Author Response · Author response to Decision Letter 1]

5 Feb 2021

Dear Dr. Wilson,

Thank you for considering a revision of our manuscript, “Emerging fungal pathogen of an invasive grass: Implications for competition with native plant species”. We appreciate the careful attention you and the reviewers have paid to our manuscript, which we believe has resulted in a stronger contribution. 

We acknowledge Reviewer 1’s concern that we did not confirm the susceptibility of two of the three native species to B. gigantea prior to the experiment. We addressed this limitation by revising the text to more clearly acknowledge that we did not test plants for infection following the experiment and by presenting all of our results in the context of experimental inoculation. In addition, to guide readers in interpreting the results of the experiment, we made a series of changes to the manuscript, which we outline below in our response to Reviewer 1.

One suggestion to address the issue of confirming Bipolaris infection was to go back to the dried biomass plant samples and attempt to isolate Bipolaris. However, the plant tissue from the experiment has all been oven-dried and stored in a laboratory for over a year. Although we can attempt to rehydrate the tissue and examine lesions for Bipolaris gigantea spores, the absence of spores would not indicate that the plants were not infected because oven-drying may have destroyed them. We also do not believe that infection of plants in a secondary experiment would confirm infection of plants in the focal experiment. However, we do reference Lane et al. 2020 in our manuscript, which used Microstegium vimineum and Elymus virginicus seeds from the same source used for our experiment, inoculated plants with B. gigantea using nearly identical methods to those used for our experiment, and reisolated B. gigantea spores from lesions on the plants. 

We designed our experiment with a control treatment and an inoculation treatment, where plants only differed in whether or not they were exposed to Bipolaris gigantea spores. We believe this design allows for interpretation within the context of B. gigantea inoculation. We acknowledge that the experiment would have been strengthened by examining leaves for spores, and potentially re-isolating Bipolaris prior to oven-drying the material. We did not perform this step for this experiment due to time limitations (this was a summer undergraduate research project), the absence of lesions on mock-inoculated plants, and our confidence in the B. gigantea isolation and inoculation method.

Thank you for your consideration,

Amy Kendig 

Dear Dr. Kendig,

Thank you for submitting your manuscript to PLOS ONE. After careful consideration, we feel that it has merit but does not fully meet PLOS ONE’s publication criteria as it currently stands. Therefore, we invite you to submit a revised version of the manuscript that addresses the points raised during the review process.

Reviewer 2 and 3 felt the paper was much improved and have recommended accept. However, reviewer 1 was not satisfied and recommended rejection. The main sticking point is still the issue with Bipolaris. I wonder if it is possible to recover Bipolaris from infected lesions, i.e. perform Koch's postulates on a small number of representative infected leaves? Or, perhaps it is possible to artificially infect some plants with Bipolaris spores? Otherwise, the authors will need to better convince reviewer 1of the merits of the study as it stands.

We look forward to receiving your revised manuscript.

Kind regards,

Richard A Wilson

Academic Editor

PLOS ONE

Reviewers' comments:

Reviewer's Responses to Questions

Comments to the Author

1. If the authors have adequately addressed your comments raised in a previous round of review and you feel that this manuscript is now acceptable for publication, you may indicate that here to bypass the “Comments to the Author” section, enter your conflict of interest statement in the “Confidential to Editor” section, and submit your "Accept" recommendation.

Reviewer #1: (No Response)

Reviewer #2: All comments have been addressed

Reviewer #3: All comments have been addressed

2. Is the manuscript technically sound, and do the data support the conclusions?

Reviewer #1: No

Reviewer #2: Yes

Reviewer #3: (No Response)

3. Has the statistical analysis been performed appropriately and rigorously?

Reviewer #1: Yes

Reviewer #2: Yes

Reviewer #3: (No Response)

4. Have the authors made all data underlying the findings in their manuscript fully available?

Reviewer #1: Yes

Reviewer #2: Yes

Reviewer #3: (No Response)

5. Is the manuscript presented in an intelligible fashion and written in standard English?

Reviewer #1: Yes

Reviewer #2: Yes

Reviewer #3: (No Response)

6. Review Comments to the Author

Reviewer #1: Major Comments

In this study, the authors address plant competition and plant pathogen infection. However, if the authors truly believe their study can stand without confirming that the native plants are susceptible to Bipolaris, then perhaps the authors should omit all text about Bipolaris infecting native plants. Currently, Line 276 says “Bipolaris gigantea inoculation resulted in lesions on all three native plant species, but only in the presence of Microstegium”. This statement cannot be made with confidence, the authors do not know if those lesions were caused by Bipolaris. Similar statements appear throughout the text (i.e L 333), and raises concern.

We understand your point but disagree. Bos and Parlevliet (1995) define inoculation as, “the ‘application of microorganisms or virus particles to a host or into a culture medium’ (28, 36), or, better, ‘the transfer of material containing a parasite or its propagules (a) into or onto tissues of an organism to initiate infection, (b) into or onto a culture medium for propagation’ (21). It is a human activity, even when vectors are used for transfer.” 

They define infection as “the ‘process or state of establishment of a pathogenic microorganism or virus in a living organism’ (2), or ‘the entry of an organism or virus into a host and the establishment of a permanent or temporary parasitic relationship’ (28, 36). NPV (21) clearly defined infection as ‘the landing of a pathogen on a host and the initiation of any type of parasitic activity.’ Infection is thus closely associated with pathogenic activity.” 

These definitions are consistent with the plant pathology literature and are the ones we applied to our study. Therefore, the phrase “Bipolaris gigantea inoculation” is the most accurate description of the experimental treatment. Plants were inoculated with B. gigantea spores suspended in sterile deionized water with 0.1% Tween 20 or with a control (sterile deionized water with 0.1% Tween 20) that lacked B. gigantea spores (Lines 153–162). 

Bos L, Parlevliet JE. Concepts and terminology on plant/pest relationships: Toward consensus in plant pathology and crop protection. Annu Rev Phytopathol. 1995;33: 69–102. doi:10.1146/annurev.py.33.090195.000441

To emphasize that we interpret the results within the context of these two treatments and to differentiate inoculation from infection, we added the following text to the Methods:

“Because we did not test leaves for infection with B. gigantea following inoculation, we present results in the context of the inoculation treatments rather than infection status.” (Lines 196–198)

In addition, we also revised the text to further emphasize that we interpreted the results within the context of inoculation rather than infection, and that lesions are an approximation of infection, by:

Changing the title to “Emerging fungal pathogen of an invasive grass: Implications for competition with native plant species”

Adding the text:

“We used these visual assessments of Bipolaris-like lesions to approximate B. gigantea infection of experimental plants.” (Lines 186–187)

Removing reference to Elymus “susceptibility” on line 280.

Adding “inoculation with” to the Discussion text: “We evaluated how inoculation with the emerging fungal pathogen B. gigantea affected the biomasses of three native species in competition with the invasive plant Microstegium.” (Lines 336–337).

Removing the reference to Elymus (as a species negatively affected by B. gigantea) in this Discussion text: “If Microstegium biomass amplifies B. gigantea incidence on native species in the field, B. gigantea could drive apparent competition between Microstegium and species negatively affected by B. gigantea [17].” (Lines 419–421)

Adding “inoculation with” to the Discussion text: “We used a greenhouse experiment to demonstrate that inoculation with a fungal leaf spot pathogen that has accumulated on a widespread invasive grass had unique effects on the growth of native species but did not modify biomass responses of native species to Microstegium density.” (Lines 460–463)

I acknowledge the authors explanation that the invasive competitor may have increased humidity (L413), which may have influenced disease symptoms on native plants, but it still seems like a bit of stretch. The authors should have confirmed that the pathogen can infect native plants, as part of a preliminary study or experimental design. This careful approach, would have allowed the authors to determine if humidity was a factor affecting susceptibility among native plants. Humidity could have easily been manipulated by confining the plant’s air-space with a plastic dome cover, or an autoclave bag, while exposing plants to Bipolaris. This approach is common in plant pathology cases that address foliar pathogens that infect via stomata. As it stands, the authors can only speculate if humidity was a factor.

We agree that it is speculation that native plants had foliar lesions in the presence of Microstegium due to changes in microclimate. As such, we included the word “perhaps” when describing this possible scenario in the Discussion, as well as an alternative scenario (transmission from Microstegium). (Lines 414–418)

Please note that we did in fact place plastic bags over all plants in the experiment for seven days following B. gigantea or mock inoculation treatments to encourage infection (Lines 162–164). 

Finally, we did not test the effects of humidity on lesion formation in this experiment explicitly, but we recognize the importance of better understanding this possible mechanism and have therefore addressed this question in recent and forthcoming growth chamber, greenhouse, and field experiments.

In conclusion, the mere fact that the authors did not determine if Bipolaris can infect the native plants raises concern. It is possible that the lesions may not be from Bipolaris. It may be possible the invasive carries a seedborne endophyte, that is asymptomatic in the invasive, but when transmitted, causes symptoms on the natives, and this endophyte may not be Bipolaris.

We recognize the reviewers concern that the lesions may not be from B. gigantea but we think this possibility is highly unlikely. Nevertheless, we have made every effort to be clear that we are interpreting the results in light of the inoculation treatment and have been careful to provide justification for our interpretations throughout the paper. 

We determined that B. gigantea can infect E. virginicus in a previous study (Lane et al. 2020, reference #35) and we acknowledge that we did not test the susceptibility of D. clandestinum and E. spectabilis to B. gigantea. We aimed to be transparent with readers about our a priori knowledge of their potential susceptibility (Lines 93–98). 

It is possible, although we think unlikely, that Microstegium transmitted a seedborne endophyte to the native plants, which caused their lesions. However, native plants only formed lesions following B. gigantea inoculation, not mock inoculation. This result would suggest that the B. gigantea inoculation increased the chances of lesions forming due to the seedborne endophyte. Because the B. gigantea inoculation treatment only differed from the mock inoculation treatment in that B. gigantea spores were present, this suggests that B. gigantea alone mediated the plant–endophyte interaction. While that may have occurred, it is far more parsimonious that B. gigantea itself caused the lesions. 

If in fact B. gigantea mediated a plant–endophyte interaction, we would expect to see a similar interaction play out in the field (Microstegium seeds were collected from the field), making our results still relevant for the ecological interactions of the species. While we do not address this specific case in the Discussion, we acknowledge the potential roles of seedborne pathogens, contamination, and co-occurrence of multiple pathogens (Lines 442–457).

We did not have sufficient data to test the effects of Microstegium density on native plant disease incidence and we describe the effects of Microstegium density on native plant lesions as trends (Lines 282–285). We do not want to mislead readers in our certainty that Microstegium is necessary for lesions to form on the native species. There were multiple cases in which Microstegium was present and lesions did not form. Therefore, we revised the Results text as follows:

“Bipolaris gigantea inoculation resulted in lesions on all three native plant species but only in some of the Microstegium density treatments (Fig 3).” rather than “…only in the presence of Microstegium” (Lines 279–280). 

Accordingly, we also revised the Discussion text:

“Fungal lesions were observed on some native plants grown with Microstegium” rather than “Fungal lesions were observed on native plants grown with Microstegium” (Lines 414–415)

To help clarify that lesions were only observed with B. gigantea inoculation and not with mock inoculation (besides the single pot we believe was erroneously inoculated; Lines 183–185), we edited the x-axis of Fig. S1 and included data from the mock inoculation treatment in Figs. 2 and 3.

Reviewer #2: (No Response)

Reviewer #3: (No Response)

7. PLOS authors have the option to publish the peer review history of their article (what does this mean?). If published, this will include your full peer review and any attached files.

Do you want your identity to be public for this peer review? For information about this choice, including consent withdrawal, please see our Privacy Policy.

Reviewer #1: No

Reviewer #2: No

Reviewer #3: No

---

## [Editor Report · Decision Letter 2]

9 Feb 2021

Emerging fungal pathogen of an invasive grass: Implications for competition with native plant species

PONE-D-20-23926R2

Dear Dr. Kendig,

We’re pleased to inform you that your manuscript has been judged scientifically suitable for publication and will be formally accepted for publication once it meets all outstanding technical requirements.

Kind regards,

Richard A Wilson

Academic Editor

PLOS ONE
---

## [Editor Report · Acceptance letter]

18 Feb 2021

PONE-D-20-23926R2 

Emerging fungal pathogen of an invasive grass: Implications for competition with native plant species 

Dear Dr. Kendig:

I'm pleased to inform you that your manuscript has been deemed suitable for publication in PLOS ONE. Congratulations! Your manuscript is now with our production department. 

Kind regards, 

on behalf of

Dr. Richard A Wilson 

Academic Editor

PLOS ONE